# FAST NEURAL ARCHITECTURE SEARCH
# WITH RANDOM NEURAL TANGENT KERNEL

## ABSTRACT

Neural architecture search (NAS) is very useful for automating the design of DNN architectures. In recent years, a number of methods for training-free NAS have been proposed, and reducing search cost has raised expectations for real-world applications. In a state-of-the-art (SOTA) training-free NAS based on theoretical background, i.e., NASI, however, the proxy for estimating the test performance of candidate architectures is based on the training error, not the generalization error. In this research, we propose a NAS based on a proxy theoretically derived from the bias-variance decomposition of the normalized generalization error, called NAS-NGE. Specifically, we propose a surrogate of the normalized 2nd order moment of Neural Tangent Kernel (NTK) and use it together with the normalized bias to construct NAS-NGE. We use NAS Benchmarks to demonstrate the effectiveness of the proposed method by comparing it to SOTA training-free NAS in a short search time.

## 1 INTRODUCTION

Deep neural networks (DNNs) continue to produce impressive results in real-world applications such as machine translation and object detection (Pouyanfar et al., 2018). For each task, researchers have developed DNNs with suitable architecture (He et al., 2016; Xie et al., 2017; Huang et al., 2017; Chen et al., 2017). With the practical application of DNNs, various research efforts are also underway to theoretically elucidate the overwhelming performance of DNNs (Allen-Zhu et al., 2019; Lu et al., 2020). One promising research result is Neural Tangent Kernel (NTK) defined from the inner product of the gradient for the DNN, which can be used to analyze generalization and convergence performance in the infinite-width limit of DNNs (Jacot et al., 2018; Lee et al., 2019). NTKs for various architectures have also been derived analytically, and the theoretical understanding of DNNs is advancing (Huang et al., 2020; Arora et al., 2019; Alemohammad et al., 2021; Yang, 2020). For any given architecture, as with feed forward network, it has also been confirmed that the NTK is kept unchanged during the learning process in the infinite-width limit (Yang & Littwin, 2021). However, actual DNNs have finite width, and state-of-the-art DNNs are often finite-width, deep-layered DNNs, and research is actively underway to bridge the gap between infinite-width DNNs and finite-width DNNs (Hanin & Nica, 2020a; Littwin et al., 2021; Hu & Huang, 2021; Li et al., 2021; Seleznova & Kutyniok, 2022b; Littwin et al., 2020; Huang & Yau, 2020). NTKs of DNN in finite width have randomness, and fluctuations in NTKs are tied to generalization performance (Geiger et al., 2020; Littwin et al., 2021).

On the other hand, the overwhelming performance of DNNs requires the effort of manually designing DNN architectures for each task. Neural Architecture Search (NAS) has been proposed to reduce the effort (Elsken et al., 2021). Although training-based NAS (Liu et al., 2019) performs well on a variety of tasks, the computational cost is considerably high because the DNN needs to be trained during the search. Therefore, a training-free NAS using a zero-cost proxy was proposed (Abdelfattah et al., 2021). This reduced the computational cost by far. However, this method is not based on theoretical understanding. Shu et al. (2022a) proposed a training-free NAS (NAS at Initialization; NASI) based on the NTK trace norm. However, NASI is based on a measure of training performance, not the generalization performance of the DNN architecture. Therefore, it is desirable to find a zero-cost proxy, with a theoretical background, that predicts the generalization performance of any DNN architecture.

To meet the above needs, we proposed a new training-free NAS method based on generalization performance. The generalization performance can be decomposed into bias and variance. We consider the generalization performance normalized by the power of the expected value of the output to *remove the scale effect of output for each architecture.* Besides, the normalization has *another effect on stabilizing computation concerning NAS.* Specifically, we construct a training-free NAS based on normalized variance to estimate the performance of candidate architectures, by using normalized NTK moment as a proxy of normalized variance. In addition, we construct a training-free NAS that uses both normalized variance and normalized bias to estimate the performance of candidate architectures, by using normalized NTK moment and trace norm as a proxy of normalized variance and bias. Next, using benchmark for studying NAS (Zela et al., 2020), we demonstrate the effectiveness of the proposed training-free NAS by comparing it to a conventional NAS in a short search time.

The main contributions are summarized as follows:

- We theoretically analysed DNN generalization with bias-variance decomposition and NTK.
- We proposed a computationally efficient training-free NAS based on the results of analysis.
- *In the short search time,* the proposed NAS outperforms state-of-the-art training-free NAS.

In the remainder of the paper, we introduce some related studies in Section 2. Section 3 describes the proposed method. Section 4 describes numerical experiments. Section 5 describes the conclusion of our research.

## 2 RELATED WORK

There have been many attempts to devise or tune DNN architectures to improve generalization performance in natural language processing (Vaswani et al., 2017), image recognition (Liu et al., 2021), and sound event detection (Wakayama & Saito, 2022). A wide range of applications of DNN pushes an effort toward NAS (Ren et al., 2021). As shown in Shu et al. (2022a), architectures found in NAS are even surpassing the performance of manually designed architectures. NAS consists of three components (Elsken et al., 2021). The first is the design of a search space containing candidate architectures. The second is a method for predicting the test performance of candidate architectures. The third is a method for determining the optimal architecture from the search space using predictive metrics. Our research focuses mainly on the second factor and proposes a computationally efficient NAS method.

### 2.1 NEURAL ARCHITECTURE SEARCH

An overview of NAS research will be presented. NAS is a technology that automatically designs DNN architectures. The first NAS by reinforcement learning, evolutionary algorithm, random search, and bayesian optimization, required full training of the parameters of candidate architectures to predict test performance. The technology used so many computational resources that it could only be used by a limited number of companies with access to a large number of computers equipped with GPUs (Ren et al., 2021; Elsken et al., 2021). Therefore, one-shot NAS, in which the parameters of candidate architectures need only be learned once, has been developed (Pham et al., 2018; Dong & Yang, 2019; Xie et al., 2019; Liu et al., 2019; Wang et al., 2023). Here, the technique shares weights among candidate architectures. One-shot NAS such as SNAS (Xie et al., 2019) and DARTS (Liu et al., 2019) probabilistically explore architectures, and DARTS possesses a cell-based search space (Shu et al., 2020; Wan et al., 2022). In addition, NAS-Bench-1Shot1 is maintained to improve reproducibility and efficiency of experiments using cell-based search space (Zela et al., 2020; Ying et al., 2019). Furthermore, Ye et al. (2022) proposes a $\beta$-DARTS to regularize the search process in DARTS. One-shot NAS is still computationally intensive for easy use, and a technique that can predict the test performance of candidate architectures in a simpler way is desired.

A new technique called zero-shot NAS, or training-free NAS, has been proposed to reduce the time required to training the model parameters in a one-shot NAS. Training-free NAS is used to evaluate candidate architectures with zero-cost proxies. The proxies, such as Grad, SNIP, GraSP, synflow, fisher and jacob cov, use gradient information and are combined with pruning and random search algorithms to construct a NAS that requires less computation time than a one-shot NAS (Abdelfattah et al., 2021). On the other hand, a limited number of proxies, such as NNGP-NAS (Park et al., 2020), NASWOT (Mellor et al., 2021), TE-NAS (Chen et al., 2021), KNAS (Xu et al., 2021), GradSign

(Zhang & Jia, 2022), ZiCo (Li et al., 2023) and NASI (Shu et al., 2022a), have also been proposed with a theoretical insight.

Park et al. (2020) proposed a NAS using NNGP-kernel as a zero-cost proxy. The NNGP-NAS mimics the behavior of reservoir computing rather than DNN itself, since NNGP-kernel assumes that DNNs are infinite width and consists only of the derivatives of the final layer. NASWOT (Mellor et al., 2021) predicts test performance using a proxy based on the overlap in activity between data in the initialized DNN. In addition, training-free NAS is achieved through random search. TE-NAS (Chen et al., 2021) uses the number of linear regions in the input space (Xiong et al., 2020) as a measure of expressivity and the condition number (ratio of the largest eigenvalue to the smallest eigenvalue) of NTK (Jacot et al., 2018) as a measure of trainability. The combination of the two measures is used to predict the test performance, and training-free NAS is achieved through search by a pruning. KNAS (Xu et al., 2021) is a method that uses as a proxy the average of all the elements of the NTK calculated using the parameters at initialization. Here, the search space is a randomly generated candidate architecture. GradSign (Zhang & Jia, 2022) is designed based on an analysis of the landscapes in the sample-wise optimization. ZiCo (Li et al., 2023) is a zero-cost proxy based on a theoretical analysis of gradient properties across different samples.

Although above six methods, i.e. NNGP-NAS, NASWOT, TE-NAS, KNAS, GradSign, and ZiCo, are proxies devised inspired by theory, the predictive performance of NAS has only been shown experimentally, and a NAS method based on a rigorous theoretical background has been desired.

NASI (Shu et al., 2022a) has been proposed that uses NTK theory to construct a zero-cost proxy and select an architecture that minimizes training errors by using the differentiable search method used in one-shot NAS. In other words, NASI is a NAS with a solid theoretical background that effectively combines zero-cost proxies and efficient search methods. Therefore, compared to a NAS based on empirical proxy (Abdelfattah et al., 2021), and a NAS by pruning (Chen et al., 2021), etc., NASI is very promising as a baseline among training-free NAS. Specifically, NASI is based on the representation of training errors such as MSE in terms of eigenvalues and eigenvectors of NTKs. NASI is attributed to the problem of minimizing the function of the mean value of eigenvalues, which is the upper bound of training errors, i.e., maximizing the trace norm of NTK (Shu et al., 2022a). The next section details the NTK theory used as a zero-cost proxy for NASI and the proposed training-free NAS.

## 2.2 NEURAL TANGENT KERNEL

In architectures used for real-world tasks such as CNNs, a DNN with a wide width corresponding to the infinite width limit of NTK performs better than typical DNNs with finite width and deep layers, when there is only a small amount of data (Arora et al., 2020). However, it has been found empirically that a general DNN with a finite width and deep layers performs better when there is a large amount of data (Lee et al., 2020).

There are a number of ongoing works to fill the gap between the generalization and convergence performance of infinite-width DNNs expressed by the NTK, and finite-width DNNs used in practice. Hanin & Nica (2020a) is the first study to analyze the properties of finite width NTKs corresponding to feedforward networks. The study points to the exponential increase in the normalized second-order moment of NTK elements with the ratio of depth to width. Subsequently, Littwin et al. (2021) derived an analytical expression of the upper and lower bounds of the normalized second-order moments of the elements of NTK corresponding to ResNet (He et al., 2016) and DenseNet (Huang et al., 2017). In addition, Littwin et al. (2020) numerically calculated the normalized second-order moments of the elements of NTK corresponding to ResNeXt (Xie et al., 2017). Furthermore, Hu & Huang (2021) analytically calculated the upper bounds of the normalized any-order moments of the elements of NTK corresponding to feedforward networks and residual networks. Geiger et al. (2020) mention that the difference between the NTK at initialization and the infinite width limit NTK is larger than the difference between the NTK at initialization and the NTK after training.

As Geiger et al. (2020) theoretically stated, the fluctuation of the random NTKs at initialization for finite-width DNNs affects the generalization error of DNN. On the other hand, the analysis in Hu & Huang (2021) indicates that the variance of NTK can be considerably large. Therefore, we use the normalized moment of NTK to stabilize the estimation and computation for statistics related to NAS.

## 3    PROPOSED METHOD

In this section, we propose a new method for NAS, which uses the normalized generalization error that takes bias and variance into account. Note that Neural Architecture Search (NAS), such as NASI (Shu et al., 2022a), chooses an architecture that speeds up the decay of training error, meaning that only bias is considered. On the other hand, our method considers both bias and variance under the squared loss. To construct a training-free NAS, we approximate the dynamics of DNN learning using the NTK. We show that the normalized generalization error of the trained DNN is expressed using the normalized 2nd-order moment of NTK.

Let us define $f(x; \theta_0) \in \mathbb{R}^m$ as the neural network model for the input $x \in \mathbb{R}^{m_0}$ with the initial weight parameter $\theta_0$. The collection of the data is denoted by $\mathcal{D} = (\mathcal{X}, \mathcal{Y})$, in which $\mathcal{X}$ is the set of $n$ input vectors, $x_1, \ldots, x_n$, and $\mathcal{Y}$ is the set of $n$ output vectors, $y_1, \ldots, y_n$. Here, $\mathcal{X}$ and $\mathcal{Y}$ are regarded as the concatenated column vectors of the data, i.e., $\mathcal{X} \in \mathbb{R}^{nm_0}, \mathcal{Y} \in \mathbb{R}^{nm}$. Let $\theta_t$ be the parameter at the $t$-th step of a learning algorithm such as the stochastic gradient descent method. For the sake of simplicity, the same notation $\theta_t$ is used for a positive real number $t$ when the update formula of the learning algorithm is approximated by a differential equation having the continuous parameter $t$. The output value of the model with $\theta_t$ at all the input data $\mathcal{X}$ is expressed by $f(\mathcal{X}; \theta_t) \in \mathbb{R}^{nm}$. In the below, the notation $f_t(x)$ is also used for $f(x; \theta_t)$.

### 3.1    BIAS-VARIANCE DECOMPOSITION OF NORMALIZED GENERALIZATION ERROR

First of all, let us define the normalized 2nd order moment. For the $m$-dimensional random variable $X = (X_1, \ldots, X_m)$, let us define the normalized 2nd order moment $Z(X)$ by

$$Z(X) := \frac{\mathbb{E}[\|X\|^2]}{\|\mathbb{E}[X]\|^2},$$

where $\|\cdot\|$ is the Euclidean norm. In the next subsection, we consider a surrogate of $Z(f_t(x))$ using the NTK. We show some property of $Z(X)$. The normalized 2nd order moment is the scale-free quantity, that is, $Z(X) = Z(cX)$ for any non-zero constant $c$. For $X = (X_1, \ldots, X_m)$, it holds that $Z(X) = \frac{\sum_i \mathbb{E}[X_i^2]}{\sum_i \mathbb{E}[X_i]^2} \leq \sum_i \frac{\mathbb{E}[X_i^2]}{\mathbb{E}[X_i]^2} \leq \sum_i Z(X_i)$. For one-dimensional random variable $X_i$, we have $Z(X_i) = \frac{\mathbb{E}[X_i^2]}{\mathbb{E}[X_i]^2} = \frac{\mathrm{Var}[X_i]}{\mathbb{E}[X_i]^2} + 1$, i.e., $Z(X_i)$ is expressed by the ratio of the variance and the mean square.

To assess the generalization performance of candidate architectures regardless of the output scale, we use the bias-variance decomposition of the normalized generalization error. The trained DNN $f_t$ depends on the initial parameter $\theta_0$ and the data $\mathcal{D}$. Suppose that the initial parameter $\theta_0$ of the model $f_0$ is randomly determined. The expected squared error of the trained model $f_t$ on a test data $x$ is defined by

$$R(x) := \mathbb{E}_{\theta_0, \mathcal{D}}[\|f_t(x) - y_{\mathrm{true}}(x)\|^2],$$

where $y_{\mathrm{true}}(x) \in \mathbb{R}^m$ is the true output function. For the classification problems, $y_{\mathrm{true}}(x)$ stands for the one-hot-vector. The expectation is taken for $\theta_0$ and $\mathcal{D}$. Let us define

$$\mathrm{Bias}(f_t(x))^2 = \|\mathbb{E}_{\theta_0, \mathcal{D}}[f_t(x)] - y_{\mathrm{true}}(x)\|^2,$$

$$\mathrm{Var}(f_t(x)) = \mathbb{E}_{\theta_0, \mathcal{D}}[\|f_t(x) - \mathbb{E}_{\theta_0, \mathcal{D}}[f_t(x)]\|^2] = \mathbb{E}_{\theta_0, \mathcal{D}}[\|f_t(x)\|^2] - \|\mathbb{E}_{\theta_0, \mathcal{D}}[f_t(x)]\|^2.$$

To remove the effect of the scale of $f_t(x)$, we consider a normalized $R(x)$. Then, the bias-variance decomposition of the normalized $R(x)$ is written as

$$\frac{R(x)}{\|\mathbb{E}_{\theta_0, \mathcal{D}}[f_t]\|^2} = \frac{\mathrm{Bias}(f_t(x))^2}{\|\mathbb{E}_{\theta_0, \mathcal{D}}[f_t]\|^2} + \frac{\mathrm{Var}(f_t(x))}{\|\mathbb{E}_{\theta_0, \mathcal{D}}[f_t]\|^2} = \frac{\mathrm{Bias}(f_t(x))^2}{\|\mathbb{E}_{\theta_0, \mathcal{D}}[f_t]\|^2} + Z(f_t(x)) - 1.$$

Note that normalization reduces the apparent value of variance and thus stabilizes the calculation. As shown in the next section, NTKs are available to analyze the normalized generalization error.

### 3.2    A SURROGATE OF NORMALIZED 2ND ORDER MOMENT

We propose a surrogate quantity of $Z(f_t(x))$. For that purpose, we derive an approximation $\widetilde{f}_t$ for $f_t$. Then, we consider a computationally tractable upper bound of $Z(\widetilde{f}_t)$ for NAS.

As shown in (Jacot et al., 2018), the NTK is defined by $\Theta_t(\mathcal{X}, \mathcal{X}) := \nabla_\theta f(\mathcal{X}; \theta_t) \nabla_\theta f(\mathcal{X}; \theta_t)^{\mathrm{T}} \in \mathbb{R}^{nm \times nm}$. The DNN dynamics is expressed by

$$\frac{\mathrm{d} f_t(\mathcal{X})}{\mathrm{d} t} = -\eta \Theta_t(\mathcal{X}, \mathcal{X}) \nabla_{f_t(\mathcal{X})} \mathcal{L},$$

where $\eta$ is a learning rate, and $\nabla_{f_t(\mathcal{X})} \mathcal{L} \in \mathbb{R}^{nm}$ is the gradient of the loss function $\mathcal{L}$ at $f_t(\mathcal{X})$ with respect to the model output. See Lee et al. (2019) for details.

In the below, we derive $\widetilde{f}_t$. The equality $f_t(x) = f_0(x) + \sum_{j=1}^{t} \Delta f_j(x)$ holds for $\Delta f_j(x) = f_j(x) - f_{j-1}(x)$. Using the NTK, $\Delta f_j(x)$ is approximated by

$$\Delta f_j(x) = -\eta \sum_i \Theta_j(x, x_i) \nabla_{f_j(x_i)} \mathcal{L}$$

$$= -\eta \sum_i \Theta_0(x, x_i) \nabla_{f_j(x_i)} \mathcal{L} - \eta \sum_i (\Theta_j(x, x_i) - \Theta_0(x, x_i)) \nabla_{f_j(x_i)} \mathcal{L} \quad (1)$$

where $\nabla_{f_j(x_i)} \mathcal{L} \in \mathbb{R}^m$ and $\Theta_j(x, x_i) \in \mathbb{R}^{m \times m}$ is the NTK defined from $f_j$. As shown in Lee et al. (2019), the change of the NTK, $\Theta_j - \Theta_0$, goes to zero as the width of the DNN model tends to infinity. Thus, we ignore $\Theta_j - \Theta_0$. Here, $x$ in the NTK is a test input, which is not available in practice. The training data is substituted instead of the test input $x$. As Littwin et al. (2021) numerically confirmed, the diagonal value $\Theta_0(x, x)$ is asymptotically the same as the off-diagonal value $\Theta_0(x, x')$ for $x \neq x'$ as the width of the neural network model tends to infinity. Though this finding has not been rigorously proved, we apply this approximation. In the below, we assume i) $\frac{1}{nt} \sum_{i,j} \nabla_{f_j(x_i)} \mathcal{L}$ is a constant vector, say $g \in \mathbb{R}^m$, and ii) the condition number of $\mathbb{E}_{\theta_0, \mathcal{D}}[\Theta_0]$ is bounded above by a constant $\kappa_0$. As an approximation of $f_t$, let us define $\widetilde{f}_t(x)$ by

$$\widetilde{f}_t(x) = f_0(x) - \eta n t \Theta_0(x', x') g,$$

where $x'$ is a training data.

Let us compute $Z(\widetilde{f}_t(x))$. In the below, $\lambda_{\max}(A)$ and $\lambda_{\min}(A)$ are the maximum and minimum eigenvalues of the matrix $A$. The diagonal sum of the square matrix $A$ is denoted by $\mathrm{Tr} A$. Since often the initialization of the network parameters $\theta_0$ is chosen by the centered Gaussian, we can assume that $\mathbb{E}_{\theta_0}[f_0(x)] = 0 \in \mathbb{R}^m$. Indeed, the expectation of the weight parameters in the last layer vanishes under the centered Gaussian initialization. Hence, we have

$$Z(\widetilde{f}_t(x)) = \frac{\mathbb{E}_{\theta_0, \mathcal{D}}[\|f_0(x) - \eta n t \Theta_0(x', x') g\|^2]}{\|\mathbb{E}_{\theta_0, \mathcal{D}}[f_0(x) - \eta n t \Theta_0(x', x') g]\|^2}$$

$$\leq \frac{2 \mathbb{E}_{\theta_0, \mathcal{D}}[\|f_0(x)\|^2]}{\eta^2 n^2 t^2 \|\mathbb{E}_{\theta_0, \mathcal{D}}[\Theta_0(x', x') g]\|^2} + \frac{2 \mathbb{E}_{\theta_0, \mathcal{D}}[\|\Theta_0(x', x') g\|^2]}{\|\mathbb{E}_{\theta_0, \mathcal{D}}[\Theta_0(x', x') g]\|^2}.$$

One can ignore the first term for a large $n$ and $t$. Using the assumption that the condition number of $\mathbb{E}_{\theta_0, \mathcal{D}}[\Theta_0(x', x')]$ is bounded above by $\kappa_0$, we derive an upper bound of the second term as follows,

$$\frac{\mathbb{E}_{\theta_0, \mathcal{D}}[\|\Theta_0(x', x') g\|^2]}{\|\mathbb{E}_{\theta_0, \mathcal{D}}[\Theta_0(x', x') g]\|^2} \leq \kappa_0^2 m^2 Z(\mathrm{Tr}\Theta_0(x', x')). \quad (2)$$

The derivation is shown in Appendix D. We use $Z(\mathrm{Tr}\Theta_0(x', x'))$ as a surrogate quantity of $Z(f_t(x))$ up to a constant factor.

In the same way as Liu et al. (2019), we introduce the distribution $p_\alpha(\mathcal{A})$ parameterized by $\alpha$ over candidate architectures. The optimal distribution over candidate architectures is expected to attain the minimum value of the expected normalized 2nd order moment. Since the width of practical DNNs is finite, the NTK with an extremely small variance should be excluded. For this reason, let us introduce a constraint $[\nu - Z(\mathrm{Tr}\Theta_0(x', x'; \mathcal{A}))]_+$ to our objective function. Also, this constraint has the role of avoiding over-reliance on training data to search for DNN architectures. Eventually, we want to minimize the expected performance of architectures sampled from $p_\alpha(\mathcal{A})$:

$$\mathbb{E}_{\mathcal{A} \sim p_\alpha} \left[ Z(\mathrm{Tr}\Theta_0(x', x'; \mathcal{A})) + \mu[\nu - Z(\mathrm{Tr}\Theta_0(x', x'; \mathcal{A}))]_+ \right], \quad (3)$$

where $\Theta_0(x', x'; \mathcal{A})$ is the NTK $\Theta_0(x', x') \in \mathbb{R}^{m \times m}$ at the training data $x'$ with the architecture $\mathcal{A}$ and $[a]_+ = \max\{a, 0\}$. The training data $x'$ is randomly selected for each step in the minimization.

**Remark 1.** *As shown in Yang & Littwin (2021), the NTK approximation applies to a wide range of DNNs. In the binary classification problem, Zhu et al. (2022) study generalization properties of NAS for DNNs with skip connections and pseudo-Lipschitz activation functions.*

### 3.3 EVALUATION OF NORMALIZED BIAS

In addition to the normalized variance, we consider the normalized bias as another performance measure.

For the MSE loss, Lee et al. (2019) derived

$$f_t(x) = \Theta_0(x, \mathcal{X})\Theta_0(\mathcal{X}, \mathcal{X})^{-1}\mathcal{Y} + f_0(x) - \Theta_0(x, \mathcal{X})\Theta_0(\mathcal{X}, \mathcal{X})^{-1}f_0(\mathcal{X}).$$

In the infinite width limit, $\Theta_0$ converge to $\Theta^*$, which is a deterministic NTK. Then, the approximate expectation is given by

$$\mathbb{E}_{\theta_0}[f_t(x)] = \Theta^*(x, \mathcal{X})\Theta^*(\mathcal{X}, \mathcal{X})^{-1}\mathcal{Y},$$

because $\mathcal{Y}$ and $\Theta^*$ do not depend on $\theta_0$ (Seleznova & Kutyniok, 2022a). The normalized bias is

$$\frac{\text{Bias}(f_t(x))^2}{\|\mathbb{E}_{\theta_0,\mathcal{D}}[f_t(x)]\|^2} = \frac{\|\mathbb{E}_{\theta_0,\mathcal{D}}[f_t(x)] - y_{\text{true}}(x)\|^2}{\|\Theta^*(x, \mathcal{X})\Theta^*(\mathcal{X}, \mathcal{X})^{-1}\mathcal{Y}\|^2}.$$

In addition, a randomly selected training data $x_i$ is utilized as a substitute for test data $x$. The average of the denominator is given by

$$\frac{1}{n}\sum_{i=1}^{n}\|\Theta^*(x_i, \mathcal{X})\Theta^*(\mathcal{X}, \mathcal{X})^{-1}\mathcal{Y}\|^2 = \frac{1}{n}\mathcal{Y}^{\text{T}}\mathcal{Y}.$$

By replacing the denominator $\|\Theta^*(x, \mathcal{X})\Theta^*(\mathcal{X}, \mathcal{X})^{-1}\mathcal{Y}\|^2$ with its mean over the training data, the normalized bias is approximated by $n(\mathcal{Y}^{\text{T}}\mathcal{Y})^{-1}\|\mathbb{E}_{\theta_0,\mathcal{D}}[f_t(x_i)] - y_{\text{true}}(x_i)\|^2$. The approximate normalized bias corresponds to the loss function of NASI (Shu et al., 2022a), i.e., $-\text{Tr}\Theta_0(\mathcal{X}, \mathcal{X}; \mathcal{A})$, where $\Theta_0(\mathcal{X}, \mathcal{X}; \mathcal{A}) \in \mathbb{R}^{nm \times nm}$ is the NTK for the training data $\mathcal{X}$ with the architecture $\mathcal{A}$.

### 3.4 NAS BASED ON NORMALIZED GENERALIZATION ERROR

As shown in the previous sections, the surrogate of the normalized bias is given by $-\text{Tr}\Theta_0(\mathcal{X}, \mathcal{X}; \mathcal{A})$, which is used in NASI. The regularization $[\text{Tr}\Theta_0(\mathcal{X}, \mathcal{X}; \mathcal{A}) - \nu']_+$ is introduced to mitigate the overfitting (Shu et al., 2022a). Furthermore, the expected normalized 2nd order moment $\tilde{Z}(f_t(x))$ over candidate architectures is replaced with Eq. 3. Eventually, our formulation is given by the following optimization problem for the architecture distribution $p_\alpha(\mathcal{A})$,

$$\min_{\alpha} \mathbb{E}_{\mathcal{A} \sim p_\alpha}\left[ -\text{Tr}\Theta_0(\mathcal{X}, \mathcal{X}; \mathcal{A}) + \mu'[\text{Tr}\Theta_0(\mathcal{X}, \mathcal{X}; \mathcal{A}) - \nu']_+ \right]$$

$$+ \gamma\mathbb{E}_{\mathcal{A} \sim p_\alpha}\left[ Z(\text{Tr}\Theta_0(x', x'; \mathcal{A})) + \mu[\nu - Z(\text{Tr}\Theta_0(x', x'; \mathcal{A}))]_+ \right], \quad (4)$$

where the scaling factor $n(\mathcal{Y}^{\text{T}}\mathcal{Y})^{-1}$ of the normalized bias is absorbed in the hyper-parameter $\gamma$.

To perform the optimization in Eq. 3 or Eq. 4 of the proposed method, we use a continuous approximation of search algorithm similar to NASI. Using Gumbel-Softmax that approximates categorical distributions by continuous distribution, we convert the architecture search into a continuous optimization problem.

In the next section, we numerically confirm the effectiveness of the proposed training-free NAS method.

## 4 EXPERIMENTS

We proposed a new training-free method using zero-cost proxy based on generalization performance to achieve a very fast NAS. In a very short search time, we validate the search effectiveness of the proposed method in the three search spaces of NAS-Bench-1Shot1 (Zela et al., 2020) on CIFAR-10, and in the search space and three datasets of NAS-Bench-201 (Dong & Yang, 2020). First of all, we describe the details of the conditions and results of the experiments using the proposed method derived in the previous section, a conventional method, i.e., NASI (Shu et al., 2022a), and a Hybrid NAS, i.e., HNAS (Shu et al., 2022b). HNAS is also NAS based on generalization performance similar to the proposed method, but it is a hybrid NAS combining a zero-cost proxy and validation performance for Bayesian optimization (Snoek et al., 2012) unlike the proposed training-free NAS.

Table 1: Information of search spaces in NAS-Bench-1Shot1 (Zela et al., 2020). The sum of the parents number of all nodes in space is chosen to be 9. Three types of search spaces were proposed by varying the determination of the number of parents each selected block has. A loose end is node whose output does not contribute to the discrete cell output.

| | | NUMBER OF PARENTS | | | | | NUMBER OF ARCHITECTURES | |
|---|---|---|---|---|---|---|---|---|
| NODE | 1 | 2 | 3 | 4 | 5 | OUTPUT | W/O LOOSE ENDS | W/ LOOSE ENDS |
| SEARCH SPACE 1 | 1 | 2 | 2 | 2 | – | 2 | $6,240$ | $2,487$ |
| SEARCH SPACE 2 | 1 | 1 | 2 | 2 | – | 3 | $29,160$ | $3,609$ |
| SEARCH SPACE 3 | 1 | 1 | 1 | 2 | 2 | 2 | $363,648$ | $24,066$ |

Table 2: Experimental results of NAS based on normalized 2nd order moment (NAS-NOM). The test error is reported along with the mean and standard error after querying results of 3 independent training on each 5 independently searched architecture, using NAS-Bench-1Shot1. In two search spaces, NAS-NOM outperforms the conventional method, i.e., NASI (Shu et al., 2022a). The running time (in second), including calculation for proxy such as NTK, is evaluated on a single Tesla V100S and reported, in all of the following experiments.

| | NASI | PROPOSED |
|---|---|---|
| STEP | 40 | 20 |
| TIME | $33.0 \pm 3.71$ [S] | $28.1 \pm 2.96$ [S] |
| SEARCH SPACE 1 | $0.096 \pm 0.054$ | $\underline{0.085 \pm 0.011}$ |
| SEARCH SPACE 2 | $\underline{0.077 \pm 0.008}$ | $0.087 \pm 0.016$ |
| SEARCH SPACE 3 | $0.099 \pm 0.039$ | $\underline{0.070 \pm 0.009}$ |

## 4.1 EXPERIMENTS ON NAS-BENCH-1SHOT1

Experiments in the NAS are computationally very expensive and it is virtually impossible to perform a proper scientific evaluation with many repeated runs to draw statistically robust conclusions (Ren et al., 2021; Elsken et al., 2021). Therefore, NAS-Bench-101 (Ying et al., 2019) was introduced to simulate an arbitrary number of runs of NAS methods inexpensively. NAS-Bench-101 is a large tabular benchmark with a unique 423k cell architecture. NAS-Bench-1Shot1 reuses NAS-Bench-101 to inexpensively benchmark NAS methods. Table 1 shows the characteristics of each search space.

We have conducted experiments using the two proposed methods and describe the experimental conditions and results in detail. The proposed method was implemented based on the code published as the supplemental materials [1] of a conventional method, i.e., NASI (Shu et al., 2022a). We compared the effectiveness between the proposed method using Eq. 3, i.e., NAS based on normalized 2nd order moment (NAS-NOM), with 20 search steps and NASI with 40 search steps. We also compared the effectiveness between the proposed method using Eq. 4, i.e., NAS based on normalized generalization error (NAS-NGE), with 20 search steps and NASI with 40 search steps. In all experiments, the search budget is comparable and very short, specifically about 30 seconds. Here, the test error for each method is reported along with the mean and standard error after querying results of 3 independent training on each 5 independently searched architecture, using NAS-Bench-1Shot1.

**Proposed : NAS based on normalized 2nd order moment (NAS-NOM)**

The number of initialized models used to calculate the normalized 2nd order moment of NTK was 4 in NAS-NOM. In addition, one data for calculating the NTK moment is randomly sampled at each step from the training data on CIFAR-10, in all of the following experiments. In all of the following experiments, the parameters of each model for calculating the NTK moment were independently initialized with Kaiming Norm Initialization (He et al., 2015). The values of the hyperparameters in Eq. 3 are as $\mu = 2.0$ and $\nu = 1.5$ for the loss of the NTK normalized 2nd order moment.

Table 2 shows that in two of these three search spaces, the proposed method, i.e., NAS-NOM, outperforms the conventional method, i.e., NASI, i.e., it usually selects neural architectures with better generalization performance.

---

[1]https://openreview.net/attachment?id=v-v1cpNNK_v&name=supplementary_material

Table 3: Experimental results of NAS based on normalized generalization error (NAS-NGE). The test error is reported along with the mean and standard error similar way to Table 2. In all three search spaces, NAS-NGE outperforms the conventional method, i.e., NASI, and the Hybrid NAS method, i.e., HNAS (Shu et al., 2022b).

| | NASI | PROPOSED | HNAS |
|---|---|---|---|
| STEP | 40 | 20 | 4 |
| TIME | $33.0 \pm 3.71$ [s] | $35.2 \pm 3.66$ [s] | $163.2 \pm 17.96$ [s] |
| SEARCH SPACE 1 | $0.096 \pm 0.054$ | $\underline{0.086 \pm 0.024}$ | $0.090 \pm 0.024$ |
| SEARCH SPACE 2 | $0.077 \pm 0.008$ | $\underline{0.074 \pm 0.006}$ | $0.092 \pm 0.021$ |
| SEARCH SPACE 3 | $0.099 \pm 0.039$ | $\underline{0.085 \pm 0.014}$ | $0.087 \pm 0.038$ |

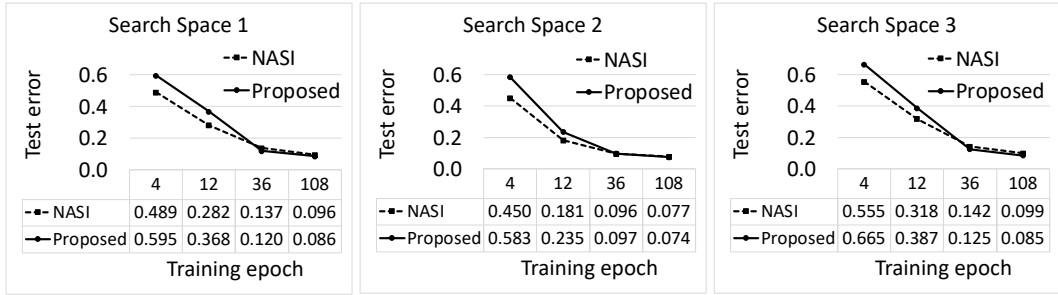

Figure 1: Results at different training epochs. The test error is reported similar way to Table 2. Compared to NASI, NAS-NGE can predict test errors attained by DNNs trained with more epochs.

**Proposed : NAS based on normalized generalization error (NAS-NGE)**

The number of initialized models used to calculate the NTK normalized 2nd order moment was 3 in NAS-NGE. On the other hand, the number of models used to calculate the NTK trace norm was 1 in NAS-NGE. Here, 64 data for calculating the NTK trace norm is randomly sampled at each step from the training data on CIFAR-10. The values of the hyperparameters in Eq. 4 are as $\mu = 2.0$ and $\nu = 1.5$ for the loss of the NTK normalized 2nd order moment. In addition, the values of the hyperparameter for the loss of the NTK trace norm in Eq. 4 is $\mu' = 1$, and another hyperparameter $\nu'$ is determined adaptively in the same way as NASI (Shu et al., 2022a). Here, the initial value of $\nu'$ is 1000. The hyperparameter $\gamma$ in Eq. 4, which links the two losses, was set to 50 so that the scale of the two losses is comparable. In Hybrid NAS, i.e. HNAS (Shu et al., 2022b), we used a combination of NTK trace norm and condition number as the proxy, and queried NAS-Bench-1Shot1 (Zela et al., 2020) for validation errors after 4 epochs of training the candidate architecture. The number of iteration was set to 4. We used the publicly available implementation of HNAS [2].

Table 3 shows that the proposed method, i.e., NAS-NGE, outperforms the conventional method, i.e., NASI, in all three search spaces, i.e., it consistently selects neural architectures with better generalization performance. And, NAS-NGE also outperforms HNAS, although the search time of NAS-NGE is much shorter than that of HNAS. Fig. 1 shows that NAS-NGE can accurately predict test errors attained by DNNs trained with more epochs, compared to NASI. The high prediction accuracy of DNN's long-term training dynamics enables us to construct a computationally efficient method for NAS. Table 4 shows that NAS-NGE attains a similar performance as NASI with about three times higher computation efficiency.

### 4.2 EXPERIMENTS ON NAS-BENCH-201

In a very short search time, we also validate the search effectiveness of the proposed training-free NAS, i.e., NAS-NGE, in the CIFAR-10 search space and three datasets, i.e., CIFAR-10, CIFAR-100, and ImageNet-16-120 (Chrabaszcz et al., 2017)), of NAS-Bench-201 (Dong & Yang, 2020). NAS-Bench-201 has provides a cell-based search space containing $15,625$ architectures and a database containing test accuracy for each architectures. Here, the test accuracy is reported along with the mean and standard error after querying results of each 5 independently searched architecture, using NAS-Bench-201.

---

[2]https://github.com/shuyao95/HNAS

Table 4: Experimental results in an extended time. The test error is reported similar way to Table 2.

| | PROPOSED | PROPOSED | NASI |
|---|---|---|---|
| STEP | 20 | 60 | 360 |
| TIME | $35.2 \pm 3.66$ [S] | $104.7 \pm 10.93$ [S] | $291.3 \pm 30.48$ [S] |
| SEARCH SPACE 1 | $0.086 \pm 0.024$ | $0.072 \pm 0.013$ | $0.068 \pm 0.004$ |
| SEARCH SPACE 2 | $0.074 \pm 0.006$ | $0.074 \pm 0.008$ | $0.071 \pm 0.003$ |
| SEARCH SPACE 3 | $0.085 \pm 0.014$ | $0.068 \pm 0.006$ | $0.074 \pm 0.013$ |

Table 5: Experimental results of the proposed training-free NAS, i.e., NAS-NGE. The test accuracy is reported along with the mean and standard error after querying results of 5 independently searched architecture, using NAS-Bench-201. In all datasets, NAS-NGE outperforms other training-free NAS, i.e., NASI, ZiCo (Li et al., 2023) and TE-NAS (Chen et al., 2021), and HNAS.

| | NASI | PROPOSED | ZICO | TE-NAS | HNAS |
|---|---|---|---|---|---|
| STEP | 40 | 15 | 2 | 2 | 2 |
| TIME | $9.6 \pm 0.49$ [S] | $9.2 \pm 0.75$ [S] | $12.0 \pm 0.63$ [S] | $24.6 \pm 0.49$ [S] | $227.0 \pm 30.17$ [S] |
| CIFAR-10 | $90.84 \pm 1.62$ | $\underline{92.16 \pm 1.13}$ | $89.92 \pm 3.15$ | $90.97 \pm 2.43$ | $90.34 \pm 1.62$ |
| CIFAR-100 | $65.51 \pm 3.27$ | $\underline{67.92 \pm 2.33}$ | $63.58 \pm 5.17$ | $65.82 \pm 4.46$ | $65.04 \pm 2.39$ |
| IN-16-120 | $35.95 \pm 3.67$ | $\underline{39.21 \pm 9.11}$ | $36.92 \pm 6.12$ | $37.84 \pm 7.49$ | $36.12 \pm 3.39$ |

We describe details of experimental conditions. We compared the effectiveness between NAS-NGE with 15 search steps and a conventional training-free NAS, i.e. NASI, with 40 search steps. The number of initialized models used to calculate the normalized 2nd order moment of NTK was 3, and the number of models used to calculate the trace norm of NTK was 1, in NAS-NGE. The search budget of NASI and NAS-NGE, is comparable and very short, specifically about 10 seconds. The values of the hyperparameters in Eq. 4 are the same as in the previous experiments.

We also compared NAS-NGE to other training-free NAS, i.e., ZiCo (Li et al., 2023) and TE-NAS (Chen et al., 2021), and HNAS. We used the publicly available implementation of ZiCo [3] and TE-NAS [4]. Purely to stabilize the calculation, we moved log function built into ZiCo on the outermost side. And then, we used the same pruning for search in ZiCo as TE-NAS. The numbers of pruning were set to 2. In HNAS, we queried NAS-Bench-201 for validation accuracies and training costs after 12 epochs of training the candidate architecture. The number of iteration was set to 2. The search budget of them is each about 10 seconds, twenty to thirty seconds, or a few minutes.

Table 5 shows that the proposed NAS outperforms NASI, ZiCo, TE-NAS and HNAS, i.e., it consistently selects neural architectures with better generalization performance.

## 5 CONCLUSION

Neural Architecture Search (NAS) can design automatically for DNN architecture (Elsken et al., 2021). Training-free NAS is efficient and has high promise for real applications (Shu et al., 2022a). But, it is based on not generalization error but training error to estimate the performance of candidate architectures. Thus, we proposed a new training-free NAS based on the bias-variance decomposition of the normalized generalization error (NAS-NGE). Specifically, we used the normalized 2nd order moment of Neural Tangent Kernel (NTK) together with the normalized bias to construct NAS-NGE. In a very short search time, the effectiveness of the proposed NAS method is demonstrated compared with the conventional NAS, using NAS Benchmarks, i.e., NAS-Bench-1Shot and NAS-Bench-201. In this study, the moment of NTKs are calculated numerically. By restricting the search space to ResNet, etc., it will be possible to efficiently compute the moments of NTKs with analytical approach such as Littwin et al. (2021); Hu & Huang (2021). Future works include applications of the proposed NAS to a transformer search space (Chen et al., 2022; Chitty-Venkata et al., 2022) and development of training-free NAS for Out-of-Distribution setting.

---

[3] https://github.com/SLDGroup/ZiCo/blob/main/ZeroShotProxy
[4] https://github.com/VITA-Group/TENAS

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

Table 6: Experimental results of NAS based on normalized 2nd order moment (NAS-NOM) without constraint calculated by 8 or 1 data. The test error is reported along with the mean and standard error similar way to Table 2. Such a simplification does not significantly affect the statistical performance.

| DATA | 8 | 1 |
|---|---|---|
| STEP | 20 | 20 |
| TIME | $154.8 \pm 16.22$ [S] | $29.5 \pm 2.85$ [S] |
| SEARCH SPACE 1 | $0.109 \pm 0.033$ | $0.115 \pm 0.011$ |
| SEARCH SPACE 2 | $0.104 \pm 0.018$ | $0.099 \pm 0.019$ |
| SEARCH SPACE 3 | $0.090 \pm 0.017$ | $0.081 \pm 0.020$ |

Table 7: Experimental results of NAS based on normalized 2nd order moment (NAS-NOM) without or with constraint. The test error is reported along with the mean and standard error similar way to Table 2. In all three search spaces, NAS-NOM w/ constraint outperforms NAS-NOM w/o constraint.

| CONSTRAINT | W/O | W/ |
|---|---|---|
| STEP | 20 | 20 |
| TIME | $29.5 \pm 2.85$ [S] | $28.1 \pm 2.96$ [S] |
| SEARCH SPACE 1 | $0.115 \pm 0.011$ | $0.085 \pm 0.011$ |
| SEARCH SPACE 2 | $0.099 \pm 0.019$ | $0.087 \pm 0.016$ |
| SEARCH SPACE 3 | $0.081 \pm 0.020$ | $0.070 \pm 0.009$ |

## A  PRELIMINARY EXPERIMENTS

### A.1  VALIDITY OF USING ONE DATA

In our manuscript, setting $x = x_i$ reduces the computational cost. We tried to compute the normalized variance using $x_1, \ldots, x_n$. However, we used only $x_i$ in our experiments, since there was no significant change in value from the normalized variance computed using only $x_i$ despite of the increased computational cost.

As shown in Table.6, some preliminary experiments showed that such a simplification does not significantly affect the statistical performance of our method.

### A.2  THE NEED TO ADD CONSTRAINT

In NAS based on normalized 2nd order moment (NAS-NOM), we compared the case without a constraint and the case with a constraint to see if a constraint is necessary.

The number of initialized models used to calculate the normalized 2nd order moment of NTK was 4 in NAS-NOM without or with constraint. In addition, one data for calculating the NTK moment without or with constraint is randomly sampled at each step from the training data on CIFAR-10, in all of the following experiments.

The values of the hyperparameters in Eq. 3 are as $\mu = 2.0$ and $\nu = 1.5$ for the loss of the NTK moment with constraint.

Table 7 shows that in all three search spaces, NAS-NOM with constraint outperforms NAS-NOM without constraint, i.e., it consistently selects neural architectures with better generalization performance.

Table 8: Experimental results of NAS based on normalized 2nd order moment (NAS-NOM) without constraint. The test error is reported along with the mean and standard error similar way to Table 2. NAS-NOM using 64 models outperforms NAS-NOM using 4 models to calculate the NTK moment.

| No. model | 4 | 64 |
|---|---|---|
| Step | 20 | 1 |
| Time | $29.5 \pm 2.85$ [s] | $28.1 \pm 1.48$ [s] |
| Search Space 1 | $0.115 \pm 0.011$ | $0.082 \pm 0.012$ |
| Search Space 2 | $0.099 \pm 0.019$ | $0.082 \pm 0.016$ |
| Search Space 3 | $0.081 \pm 0.020$ | $0.076 \pm 0.008$ |

## B  ADDITIONAL EXPERIMENTS

### B.1  EXPERIMENTS ON NAS-BENCH-1SHOT1

We describe a comparison of the performance of the NAS based on normalized 2nd order moment (NAS-NOM) with different numbers of initialized models for calculating the normalized 2nd order moment of NTK.

As an additional experiment, the number of initialized models used to compute the NTK moments is increased from 4 to 64 while keeping the computation time the same about 30 seconds to verify the effectiveness of the proposed method, i.e., NAS-NOM.

Table 8 shows that in all three search spaces, computing the NTK moment of NAS-NOM with 64 initialized models compared with 4 initialized models is consistently selects neural architecture with better generalization performance.

The above results show that even when running NAS using only the NTK moment, increasing the number of models can improve the performance of zero-cost proxy, even without a constraint term. However, to consistently outperform the test performance in all search spaces, it is best to use a NAS based on normalized generalization error (NAS-NGE).

### B.2  EXPERIMENTS ON NAS-BENCH-201

In a very short search time, we also validate the search effectiveness of the proposed method, i.e., NAS based on normalized generalization error (NAS-NGE), in the CIFAR-100 search space of NAS-Bench-201 (Dong & Yang, 2020).

The search budget of NASI (Shu et al., 2022a), NAS-NGE, ZiCo (Li et al., 2023) is each about 10 seconds, that of TE-NAS (Chen et al., 2021) is about 20 seconds, and that of HNAS (Shu et al., 2022b) is about 10 minutes.

Table 9 show that the proposed training-free NAS, i.e., NAS-NGE, outperforms NASI, ZiCo, TE-NAS and HNAS, i.e., it consistently selects neural architectures with better generalization performance.

In a very short search time, we also validate the search effectiveness of the proposed method, i.e. NAS based on normalized generalization error (NAS-NGE), in the ImageNet-16-120 (Chrabaszcz et al., 2017) search space of NAS-Bench-201.

The search budget of NASI, NAS-NGE, ZiCo is each about 10 seconds, that of TE-NAS is about 20 seconds, and that of HNAS is about tens of minutes.

Table 10 show that the proposed training-free NAS, i.e., NAS-NGE outperforms NASI, ZiCo, TE-NAS and HNAS, i.e., it consistently selects neural architectures with better generalization performance.

Table 9: Results of NAS based on normalized generalization error (NAS-NGE). The test accuracy is reported along with the mean and standard error similar way to Table 5, by using CIFAR-100 on NAS-Bench-201. NAS-NGE outperforms NASI (Shu et al., 2022a), ZiCo (Li et al., 2023), TE-NAS (Chen et al., 2021) and HNAS (Shu et al., 2022b).

|  | NASI | PROPOSED | ZICO | TE-NAS | HNAS |
|---|---|---|---|---|---|
| STEP | 40 | 15 | 2 | 2 | 2 |
| TIME | $10.0 \pm 0.00$ [S] | $9.4 \pm 0.49$ [S] | $13.2 \pm 0.75$ [S] | $22.8 \pm 0.40$ [S] | $619.6 \pm 59.86$ [S] |
| CIFAR-100 | $46.52 \pm 25.57$ | $\underline{67.79 \pm 2.12}$ | $65.86 \pm 6.21$ | $55.97 \pm 27.50$ | $65.04 \pm 2.39$ |

Table 10: Results of NAS based on normalized generalization error (NAS-NGE). The test accuracy is reported along with the mean and standard error similar way to Table 5, using ImageNet-16-120 on NAS-Bench-201. NAS-NGE outperforms NASI, ZiCo, TE-NAS and HNAS.

|  | NASI | PROPOSED | ZICO | TE-NAS | HNAS |
|---|---|---|---|---|---|
| STEP | 40 | 15 | 2 | 2 | 2 |
| TIME | $7.0 \pm 0.00$ [S] | $9.6 \pm 0.49$ [S] | $12.2 \pm 0.98$ [S] | $16.8 \pm 0.40$ [S] | $1472.6 \pm 180.30$ [S] |
| IN-16-120 | $33.71 \pm 11.11$ | $\underline{39.25 \pm 2.63}$ | $37.48 \pm 6.02$ | $37.70 \pm 3.93$ | $36.12 \pm 3.39$ |

## C FURTHER SURVEY

### C.1 DNN ARCHITECTURE

Specifically, SOTA architectures are often developed in the field of natural language processing, where large amounts of data can be acquired, and a great deal of effort is expended by developers (Vaswani et al., 2017). Architectures developed in the field of natural language processing have been applied to the field of image processing, and many researchers have continued to improve them by incorporating inductive bias in images (Liu et al., 2021). Recently, these architectures have been introduced in the field of acoustics, and the scope of DNN development has expanded significantly (Wakayama & Saito, 2022). Thus, the impact of a research in training-free NAS could be significant.

### C.2 EXTENSION OF NTK

The theoretical understanding of deep learning has attracted the interest of many researchers and practitioners (Hanin, 2018; Hanin & Nica, 2020b). Recently, NTK was proposed as a promising method to analyze convergence performance and generalization performance by making the objective function convex by introducing the restriction that the width of DNN is infinite or sufficiently wide (Jacot et al., 2018; Lee et al., 2019). NTK takes into account the gradient flow (Jacot et al., 2018). Furthermore, in the infinite width limit, the initial value of NTK and the NTK after training coincide by linearizing the function of the DNN using Taylor expansion. Hence, the DNN can be represented by kernel ridge regression with NTKs (Lee et al., 2019). NTKs for many types of architectures were derived, including CNN, ResNet and RNN, as well as NTK for Feed Forward Network, and the generalization and convergence performance of each architecture was analyzed (Huang et al., 2020; Arora et al., 2019; Alemohammad et al., 2021). For any given architecture, as with feed forward network, it has also been confirmed that the NTK is kept unchanged during the learning process in the infinite-width limit (Yang & Littwin, 2021).

Huang & Yau (2020) proposed the higher-order extension of NTK called Neural Tangent Hierarchy (NTH) to describe the dynamics of finite-width DNNs. In addition, Fort et al. (2020) studied the relationship between the time evolution of NTK and the geometry of the loss landscape. Recently, Mok et al. (2022) empirically investigated the time evolution of NTKs by showing the relationship between NTK-based indicators (F-Norm, Mean, Negative Condition Number; NGN, Label-Gradient Alignment; LGA) and different DNN architectures. Based on this survey, a NAS based on the LGA has been proposed, but it assumes learning over several epochs and is not efficient.

## D  UPPER BOUND OF THE SECOND-ORDER MOMENT

The upper bound in (2) is derived as follows.

$$
\begin{aligned}
\frac{\mathbb{E}_{\theta_0,\mathcal{D}}[\|\Theta_0(x',x')g\|^2]}{\|\mathbb{E}_{\theta_0,\mathcal{D}}[\Theta_0(x',x')g]\|^2} &\leq \frac{\lambda_{\max}(\mathbb{E}_{\theta_0,\mathcal{D}}[\Theta_0(x',x')^2])}{\lambda_{\min}(\mathbb{E}_{\theta_0,\mathcal{D}}[\Theta_0(x',x')]^2)} \\
&\leq \kappa_0^2 \frac{\lambda_{\max}(\mathbb{E}_{\theta_0,\mathcal{D}}[\Theta_0(x',x')^2])}{\lambda_{\max}(\mathbb{E}_{\theta_0,\mathcal{D}}[\Theta_0(x',x')]^2)} \\
&\leq \kappa_0^2 \frac{\mathbb{E}_{\theta_0,\mathcal{D}}[(\mathrm{Tr}\Theta_0(x',x'))^2]}{(\frac{1}{m}\mathbb{E}_{\theta_0,\mathcal{D}}[\mathrm{Tr}\Theta_0(x',x')])^2} \\
&= \kappa_0^2 m^2 Z(\mathrm{Tr}\Theta_0(x',x')).
\end{aligned}
$$

The numerator in the third inequality is given by $\lambda_{\max}(\mathbb{E}[\Theta_0^2]) \leq \mathbb{E}[\lambda_{\max}(\Theta_0^2)] \leq \mathbb{E}[(\mathrm{Tr}\Theta_0)^2]$. In order to reduce the computation cost, the maximum eigenvalue in the above inequalities is bounded using the trace such as $\frac{1}{m}\mathrm{Tr}A \leq \lambda_{\max}(A) \leq \mathrm{Tr}A$ for a positive semi-definite matrix $A \in \mathbb{R}^{m \times m}$.

## E  DETAILS OF NAS-BENCH-1SHOT1

Three types of search spaces were proposed by varying the determination of the number of parents each selected block has. Here, the sum of the parents number of all nodes in the search space is chosen to be 9. Each search space has a different number of architectures with or without loose ends, as shown in Table 1. A loose end is node whose output does not contribute to the discrete cell output.

Given weights of the cell architecture, the testing error and the validation error can be queried from NAS-Bench-101 as follows. First, in each *choice block* (Zela et al., 2020), an operation with the highest architecture weight is selected to determine the operation. Second, the parents of each choice block and output are determined by choosing the top-k edges. Third, the list of operations is constructed from the first procedure, the cell adjacency matrix is constructed from the second procedure, and these are used to query NAS-Bench-101 for the testing and validation errors.

## F  EXPERIMENTS IN DARTS SEARCH SPACE

In numerical experiments in Section 4, we confirmed that our method efficiently works under less than about one minute of computation. Here, let us validate the search effectiveness of NAS-NGE with a more computation time using DARTS search space (Shu et al., 2022a), which is larger than NAS Benchmarks. The values of the hyperparameter in Eq. 4 is $\mu' = 2$, and $\nu'$ is determined adaptively, where the initial value of $\nu'$ is 500. The values of $\mu$, $\nu$ and $\gamma$ are set to same in previous experiments (Sec. 4.1). Table 11 shows that NAS-NGE attains a similar performance as NASI with about 1.5 times higher computation efficiency in DARTS search space.

Table 11: Experimental results in the DARTS search space. The test accuracy is reported along with the mean and standard error after 600 epochs training on each 3 independently searched architecture.

|  | NASI | PROPOSED |
|---|---|---|
| STEP | 20 | 10 |
| TIME | $599.0 \pm 0.00$ [S] | $392.7 \pm 0.47$ [S] |
| CIFAR-10 | $96.85 \pm 0.23$ | $97.02 \pm 0.07$ |

