# OpenReview forum: "Fast Neural Architecture Search with Random Neural Tangent Kernel"
_ICLR.cc/2024/Conference — Submitted to ICLR 2024_

### Official Review · Reviewer_6Yvs · 2023-10-19

**Soundness:** 3 good
**Presentation:** 3 good
**Contribution:** 3 good
**Rating:** 6
**Confidence:** 3

**Summary:**

Neural Architecture Search (NAS) automates DNN architecture design. Existing training-free NAS methods aim to reduce search costs but often rely on training error, not generalization error. We propose NAS-NGE, using bias-variance decomposition of the normalized generalization error, outperforming SOTA training-free NAS in short search times with NAS Benchmarks.

**Strengths:**

The paper is readable and well-written. It demonstrates a clear sense of purpose, emphasizing the importance of focusing on generalization performance rather than training performance when conducting NTK-based NAS. The paper then provides a mathematical framework for this approach and demonstrates it effectively. The overall structure, from the statement of purpose to the specific approach, is well organized. The experiments also report that the proposed approach often outperforms existing methods.

**Weaknesses:**

I'm not sure about the significance of the contribution. Although the experimental results appear promising, the derivation of the bias-variance decomposition and its associated equations (Eq. 3 and Eq. 4) seems relatively straightforward, which could imply that the contribution is somewhat incremental to previous works. If the novelty of what you're doing can be highlighted even more, I believe it has the potential.

I believe it's important to carefully explain how much the behavior of models trained outside the NTK-regime can be accounted for by theoretical analysis using NTK, as unrealistic settings like lazy training are implicitly imposed. Currently, only the performance of NAS is reported, but a more direct analysis would increase the level of confidence. Specific points are listed in the "Questions" section.

**Questions:**

1: I believe that NAS-Bench stores performance at multiple checkpoint epochs (e.g., 4, 12, 36, 108 epochs). In this paper, empirical evaluation is conducted with the shortest training time, both for NAS-Bench-101 and NAS-Bench-201. Do you think that extending these checkpoint epochs significantly impacts the results? Do you have results under such settings? Since NTK theory assumes that NTK does not change during training, I imagine that as training epochs increase, it may deviate somewhat from the theory of infinite width. I'm interested in understanding this deviation. If possible, demonstrating its superiority based on differences in trends on epochs compared to other methods using NTKs could potentially strengthen the paper.

2: Like the TE-NAS paper (Chen et al. (2021)), can you visualize a scatter plot of the current metrics against the actual generalization performance? This might be a more direct way to validate the effectiveness compared to conducting NAS.

---

> ### Author Response · Authors · 2023-11-20
> **Official Comment by Authors**
>
> We thank the reviewers for their feedback and we appreciate the questions in the reviews.
>
> ### **The novelty of the paper**
>
> The main purpose of our paper is to propose a training-free NAS method that searches appropriate DNN architectures in a very short time compared to existing methods. For that purpose, we use the normalized generalization error, which is efficiently approximated using the normalized 2nd-order moment of NTK. The performance measure based on Eq.3 and Eq.4 is derived by combining some existing theoretical techniques of NTK. ***As shown in Figure 1 in the revised paper, however, our method, NAS-NGE, provides a highly accurate prediction of DNN's long-term training dynamics. Due to that property, we can construct a computationally efficient method for training-free NAS.***
>
>
> ### **Q. Comparison of trends on epochs to other methods using NTKs.**
>
> As the reviewer pointed out, there is a gap between NTK theory and practical performance. In the revised paper, we compare the performance of NTK-based training-free NAS methods, NASI, and our method, NAS-NGE. ***Figure 1 in the revised manuscript shows the test errors at several checkpoint epochs.*** Our numerical experiments indicate that our method, NAS-NGE, well predicts the generalization performance than NASI.

---

> > ### Comment · Reviewer_6Yvs · 2023-12-03
> >
> > Thank you for the update. I appreciate the additional experiments such as results with various epochs. I feel these enhancements have improved the understanding of the characteristics of the proposed method.
> >
> > I will keep my original rating. Please accept my apologies for the delayed response.

---

### Official Review · Reviewer_vdk2 · 2023-10-19

**Soundness:** 2 fair
**Presentation:** 2 fair
**Contribution:** 2 fair
**Rating:** 3
**Confidence:** 4

**Summary:**

The paper concerns train-free Neural Architecture Search (NAS). This concerns the task of automatically determining the architecture for a prescribed task, i.e. image classification in this case. In this paper, a core stated motivation concerns devising a NAS method with theoretical guarantees based on the generalization error. The bias-variance decomposition analysis of the normalized generalization error is used for NAS. Concretely, they use the (normalized) 2nd order moment of the NTK along with the normalized bias. The standard benchmark of Nasbench-201 is used for the comparison.

**Strengths:**

Automatically designing architectures is an important task and currently it is very costly. The train-free methods that can indeed reduce the cost of search have yet to demonstrate the capacity of DARTS or more recent models. At the same time, theoretical understanding of architectures and their inductive bias is important, therefore I do consider that the broader area is important and relevant for the ICLR audience. The analysis of the moments of NTK conducted in this work is something I have not seen before. Having said that, I am not sure of the precise conditions that this analysis holds.

**Weaknesses:**

- A strong motivation is built upon the premise that generalization guarantees are missing in train-free NAS. However, the papers Generalization guarantees for neural architecture search with train-validation split (ICML'21), Generalization Properties of NAS under Activation and Skip Connection Search (NeurIPS’22) are precisely providing generalization guarantees for different cases. Given those papers, I believe one of the core motivations is less clear to me. Could the authors elaborate on the differences from those?

- The experimental validation seems rather weak, using the dated nas-bench-201 as the main comparison platform. If the method scales so well in terms of time, why not use benchmarks with larger search space?

- The paper skips many important details and intuitions that make it quite hard to understand. For instance, why is the second-order moment used in the analysis and why is the NTK assumed here?

**Questions:**

- It seems the reported accuracies in this work are lower than existing standard networks, e.g. ResNets. Even if we do not account for those cases, the aforementioned Neurips’22 paper on the same benchmark reports results closer to standard accuracies on cifar10/100. Could the authors elaborate on those discrepancies?

- Given that one of the main claims is built around the generalization error, I would expect to see how the proposed model extends beyond the test data of the dataset, which is one of the critical points in NAS overall.

- One of the messages repeatedly relayed throughout the manuscript is the speed of the proposed method. However, I find that the reporting of “about 10 sec.” to be quite rough; could the authors conduct a refined analysis on this?

- Continuing on the aforementioned point, does the reported time include the time to compute the NTK of each architecture?

- In condition 3.2 it mentions the NTK, however the papers cited mostly focus on Relu-nets, so are there any additional conditions that should hold for the neural network in order for this analysis to hold?

- One of the sentences in the first paragraph of the introduction claims that “Architectures found in NAS are even surpassing the performance of manually designed architectures”. Is there a reference for this? How does the proposed method perform in imagenet that is the standard benchmark in image classification?

---

> ### Author Response · Authors · 2023-11-20
> **Official Comment by Authors (1/2)**
>
> We thank the reviewers for their feedback and we appreciate the questions in the reviews.
>
> ### **Our motivation and differences from [1] and [2]**
>
> In the paper [1], the authors derived an end-to-end generalization bound for DNN learning using training-validation split, and used the bound to training-based method. The proposed method is not training-free NAS. That is the main difference from our method.
>
> In the paper [2], mainly the authors consider theoretical analysis of training-free NAS methods for binary classification problems. On the other hand, we proposed a NTK-based approximation of the normalized generalization error for regression and classification problems. We find that our method provides an accurate prediction of DNN’s long-term training dynamics. Due to that property, we can construct a computationally efficient method for training-free NAS. ***We added numerical experiments in Figure 1 of the revised paper.***
>
>
>
> ### **Why is the second-order moment used in the analysis and why is the NTK assumed here?**
>
> The second-order moments naturally appear since we consider the squared error as the loss. NTK is used to approximate the learning dynamics of DNN. By approximating learning dynamics using NTK, we can estimate the generalization performance of DNNs after several learning steps without learning. We added the details in the first paragraph of Section 3.
>
>
> ### **Q1. Discrepancies from benchmark reports in [2]**
>
> Our numerical studies indicate that NAS-NGE can select an appropriate DNN architecture in a very short time. ***Table 4 in the revised paper shows numerical results using NAS-Bench-1shot1, which has a larger search space than NAS-bench-201.*** We see that NAS-NGE is about three times more computationally efficient than NASI and attains a comparable accuracy.
>
>
> ### **Q2. how does the proposed model extend beyond the test data of the dataset.**
>
> If the reviewer asks about the relation of NAS to out-of-distribution (OOD) generalization, that’s a very interesting problem. Thank you for the suggestion. ***The theoretical evaluation of NAS under the OOD setting was added to the Conclusion as an interesting future work.***
>
> ### **Q3. a refined analysis on reporting computation cost**
>
> In numerical experiments, the accurate computation time was supplemented in the tables.
>
> ### **Q4. the reported time include the time to compute the NTK of each architecture?**
>
> Yes, the computation time of NTK is included. This is supplemented in the caption of Table 2.
>
> ### **Q5. the papers cited mostly focus on ReLU-nets, so are there any additional conditions?**
>
> As shown in [3], the NTK approximation applies to a wide range of DNNs. In the binary classification problem, Zhu et al.[2] studied generalization properties of NAS for DNNs with skip connections and pseudo-Lipschitz activation functions. ***Description on these works were added to Remark 1 in the end of Section 3.2 of the revised paper.***
>
> ### **Q6. "Architectures found in NAS are even surpassing the performance of manually designed architectures". Is there a reference for this?**
>
> The paper [4] commented as follows: "In the literature, various search algorithms (Luo et al., 2018; Zoph et al., 2018; Real et al., 2019) have been proposed to search for architectures with comparable or even better performance than the handcrafted ones." We referred to [4] in the first paragraph of Section 2.
>
>
>
> ### **References**
>
> [1] S. Oymak, M. Li, M. Soltanolkotabi, "Generalization Guarantees for Neural Architecture Search with Train-Validation Split", ICML 2021
>
> [2] Z. Zhu, F. Liu, G. Chrysos, V. Cevher, "Generalization Properties of NAS under Activation and Skip Connection Search", NeurIPS 2022.
>
> [3] G. Yang and E. Littwin, "Tensor programs iib: Architectural universality of neural tangent kernel training dynamics", ICML 2021.
>
> [4] Y. Shu, et al., "NASI: Label- and data-agnostic neural architecture search at initialization", ICLR 2022.

---

> ### Author Response · Authors · 2023-11-23
> **Official Comment by Authors (2/2)**
>
> We supplemented numerical experiments with a larger search space.
>
> > If the method scales so well in terms of time, why not use benchmarks with larger search space?
>
> ***In the revision, we added numerical experiments with the DARTS search space in Table 11 of Appendix F.*** The result is shown in the following table. In the experiment, NASI and the proposed method with about 10 minutes of computation were applied for NAS. Our method attains a comparable prediction performance with about 1.5 times higher computation efficiency for NAS. We hope the above result addresses the reviewer's concern.
>
> ||||
> |:---:|:---:|:---:|
> |       |NASI|PROPOSED|
> | step | 20                 |    10               |
> | time  | 599.0 $\pm$ 0.00[s]| 392.7 $\pm$ 0.47[s]|
> | test acc.      | 96.85 $\pm$ 0.23   | 97.02 $\pm$ 0.07    |
> ||||

---

### Official Review · Reviewer_C4Cd · 2023-10-31

**Soundness:** 3 good
**Presentation:** 3 good
**Contribution:** 2 fair
**Rating:** 6
**Confidence:** 4

**Summary:**

This paper proposed a novel surrogate of the generalization error of neural networks for training-free Neural Architecture Search(NAS). By normalize the estimation of generalization error on different data samples and random initialization, proposed NAS-NGE outperform the other training-free pruning based NAS methods on NAS-Bench-1Shot1, and NAS-Bench-201.

**Strengths:**

The authors analyzed the output of the model by decomposing it into bias and variance. As far as I know, this approach is new and novel.
In particular, the authors' method of normalizing the variation in output due to random initialization to provide a more reliable measure of generalization error is reasonable and well explained in the paper.

**Weaknesses:**

The purpose of NAS is to find the optimal architecture within a reasonable cost. The architecture found by the proposed method in the experiments seems to be far from optimal (even considering that it was explored in a very short time). For example, the optimal architectures in NAS-Bench-201 had accuracies of 94.37, 73.51, and 47.31, respectively, and many training-free methods discover architectures with only 1-2% accuracy difference from the optimal architectures in less than an hour [1]. The proposed method explored architectures that lagged behind the optimal architecture by 2-8% in less than a minute of search time. I am not sure how this result can be utilized. This concern would be alleviated if the authors reported the variation in the accuracy of the found architectures for different search times, or if they indicated how good the found architectures were within the overall architecture pool. It would also be helpful to compare the proposed method with other methods for training-free NAS (NASWOT[2], KNAS[3]), or to compare it with other (more expensive) NAS algorithms.

[1] Shu, Y., Cai, S., Dai, Z., Ooi, B. C., & Low, B. K. H. (2021). Nasi: Label-and data-agnostic neural architecture search at initialization. arXiv preprint arXiv:2109.00817.

[2] Mellor, J., Turner, J., Storkey, A., & Crowley, E. J. (2021, July). Neural architecture search without training. In International Conference on Machine Learning (pp. 7588-7598). PMLR.

[3] Xu, J., Zhao, L., Lin, J., Gao, R., Sun, X., & Yang, H. (2021, July). KNAS: green neural architecture search. In International Conference on Machine Learning (pp. 11613-11625). PMLR.

**Questions:**

It seems that the training set is used to calculate the NTK, why is it acceptable to calculate it with the training set alone without using the validation set?

---

> ### Author Response · Authors · 2023-11-20
> **Official Comment by Authors**
>
> We thank the reviewers for their feedback and appreciate the questions in the reviews.
>
> ### **The variation in the accuracy of the found architectures for different search times.**
>
> We conducted additional numerical experiments with NAS-Bench-1shot1 to investigate the performance of NAS in different search times. The computation time and test errors are shown in the following table. The proposed method, NAS-NGE, shows a similar search performance to NASI, with about three times higher calculation efficiency. ***The following result was added to Table 4 in the revised paper.***
>
> ||PROPOSED|PROPOSED|NASI [1]|
> |:---:|:---:|:---:|:---:|
> |STEP           |           20       |          60           |         360         |
> |TIME           |  35.2 $\pm$ 3.66[s]|  104.7 $\pm$ 10.93[s] | 291.3 $\pm$ 30.48[s]|
> |SEARCH SPACE 1 | 0.086 $\pm$ 0.024  |  0.072 $\pm$ 0.013    | 0.068 $\pm$ 0.004   |
> |SEARCH SPACE 2 | 0.074 $\pm$ 0.006  |  0.074 $\pm$ 0.008    | 0.071 $\pm$ 0.003   |
> |SEARCH SPACE 3 | 0.085 $\pm$ 0.014  |  0.068 $\pm$ 0.006    | 0.074 $\pm$ 0.013   |
>
>
> ### **Q1. Calculation of NTK with the training set alone without using the validation set.**
>
> In the paper [1], also the validation dataset is not employed. The following is the comments in [1]; "Since both theoretical (Mohri et al., 2018) and empirical (Hardt et al., 2016) justifications in the literature have shown that training and validation loss are generally highly related, we simply use $L_{\mathrm{t}}$ to approximate $L_{\mathrm{val}}$", where $L_{\mathrm{t}}$ (resp. $L_{\mathrm{val}}$) is the loss over the training (resp. validation) dataset. For the same reason, we used the training set to calculate the NTK.
>
>
>
> ### **References**
>
> [1] Y. Shu, et al., "NASI: Label- and data-agnostic neural architecture search at initialization", ICLR 2022.

---

### Author Response · Authors · 2023-11-20
**Official Comment by Authors**

Dear Reviewers,

We appreciate the valuable feedback received for our submitted manuscript. We have devoted substantial effort to thoughtfully addressing reviewers' concerns. The revised version presents a significantly improved submission that effectively tackles each reviewer's concerns (detailed individual responses can be found in our responses on OpenReview), which we hope will be reflected in a final assessment. Please let us know if anything in our response requires further clarification or discussion.

This updated version of the paper is now available.

Kind regards,

The Authors

---

### Meta-Review · Area_Chair_37WN · 2023-12-15

**Metareview:**

The authors propose a neural architecture search method guided by neural tangent kernel theory. While the idea of using NTK to guide NAS is interesting and shows some promise, this paper has several critical weaknesses:

1) The differences from prior work are not sufficiently addressed.

2) The experimental validation is limited. Key recent baselines like KNAS and Eigen-NAS are not included. Additionally, some results appear inconsistent with prior work like ZiCO which shows superior performance to what is reported here.

3) The baseline comparisons in general seem weak, lacking comparisons on complex datasets like ImageNet that other recent methods include.

4) There is reliance on an unproven approximation that raises concerns about rigor.

I would encourage the authors to consider a major revision, and the paper could be much stronger if these concerns could be properly addressed.

**Justification For Why Not Higher Score:**

N/A

**Justification For Why Not Lower Score:**

N/A

---

### Decision · Program_Chairs · 2024-01-16

Reject